# Prime editing-mediated correction of the *CFTR* W1282X mutation in iPSCs and derived airway epithelial cells

Chao Li[1][�], Zhong Liu[1][�], Justin Anderson[2,3], Zhongyu Liu[3], Liping Tang[3,4], Yao Li[3,4], Ning Peng[3,4], Jianguo Chen[3,4], Xueming Liu[5], Lianwu Fu[1,3], Tim M. Townes[1], Steven M. Rowe[3,4], David M. Bedwell[1,3], Jennifer Guimbellot[2,3], Rui Zhao[1,3]*

**1** Department of Biochemistry and Molecular Genetics, School of Medicine, University of Alabama at Birmingham, Birmingham, Alabama, United States of America, **2** Department of Pediatrics, School of Medicine, University of Alabama at Birmingham, Birmingham, Alabama, United States of America, **3** Gregory Fleming James Cystic Fibrosis Research Center, School of Medicine, University of Alabama at Birmingham, Birmingham, Alabama, United States of America, **4** Department of Medicine, School of Medicine, University of Alabama at Birmingham, Birmingham, Alabama, United States of America, **5** Key Laboratory of Imaging Processing and Intelligent Control, School of Artificial Intelligence and Automation, Huazhong University of Science and Technology, Wuhan, Hubei, China

☉ These authors contributed equally to this work.
* ruizhao@uab.edu

**Data Availability Statement:** All relevant data are within the paper and its Supporting Information files.

## Abstract

A major unmet need in the cystic fibrosis (CF) therapeutic landscape is the lack of effective treatments for nonsense *CFTR* mutations, which affect approximately 10% of CF patients. Correction of nonsense *CFTR* mutations via genomic editing represents a promising therapeutic approach. In this study, we tested whether prime editing, a novel CRISPR-based genomic editing method, can be a potential therapeutic modality to correct nonsense *CFTR* mutations. We generated iPSCs from a CF patient homozygous for the *CFTR* W1282X mutation. We demonstrated that prime editing corrected one mutant allele in iPSCs, which effectively restored CFTR function in iPSC-derived airway epithelial cells and organoids. We further demonstrated that prime editing may directly repair mutations in iPSC-derived airway epithelial cells when the prime editing machinery is efficiently delivered by helper-dependent adenovirus (HDAd). Together, our data demonstrated that prime editing may potentially be applied to correct *CFTR* mutations such as W1282X.

## Introduction

Cystic fibrosis (CF) is caused by recessive mutations in the cystic fibrosis transmembrane conductance regulator (*CFTR*) gene, which encodes a phosphorylation-regulated chloride/bicarbonate channel localized on the apical epithelial surface of the pulmonary and gastrointestinal tracts, pancreatic ducts, and the male reproductive ducts [1, 2]. Among the 2,000 mutations and genetic variations of *CFTR*, approximately 300 are disease-causing [3]. F508del, the most prevalent *CFTR* mutation, accounts for 70% of all mutant alleles. Nonsense *CFTR* mutations

**Funding:** R.Z. is supported by Cystic Fibrosis Foundation Research Grant ZHAO19G0, UAB Cystic Fibrosis Research Center Pilot Grant (ROWE15R0), and NIH R01OD026594. The funders had no role in study design, data collection and analysis, decision to publish, or preparation of the manuscript.

**Competing interests:** I have read the journal's policy and the authors of this manuscript have the following competing interests: S.M.R. provided consulting services and received grants from Novartis, TranslateBio, Galapagos/Abbvie, Synedgen/Synspira, Eloxx, Vertex Pharmaceuticals, Ionis, Astra Zenica, Renovion, Cystetic Medicines, and Arcturus. S.M.R. is the inventor or co-inventor of several patents and held stock or stock options of Synedgen/Synspira and Renovion. This does not alter our adherence to PLOS ONE policies on sharing data and materials.

affect approximately 10% of all CF patients [4]. Tremendous progress has been made in developing small-molecule drugs to treat CF. Ivacaftor (VX-770), a drug that potentiates wild-type CFTR function and mutants such as the G551D with gating defects, can benefit approximately 5% of CF patients [5–8]. Trikafta, which combines ivacaftor and two corrector drugs elexacaftor/tezacaftor (i.e., VX-445/VX-661) can benefit patients with the F508del mutation [9–14]. Despite all the progress, treatments for nonsense *CFTR* mutations are currently unavailable.

Correction of nonsense *CFTR* mutations via genomic editing represents a promising approach toward disease amelioration and improved quality of life for patients. Conventional genomic editing methods, such as CRISPR/Cas9-mediated homology-directed repair (HDR), rely on the generation of DNA double-strand breaks (DSBs) at target genes, which often introduce unwanted *de novo* mutations (insertions/deletions (INDELs)) and chromosomal rearrangements [15–19]. Furthermore, HDR requires the use of a DNA correction template in addition to the guide RNA (gRNA) and Cas9 enzyme, which makes the *in vivo* delivery of the HDR repair machinery significantly more challenging. These difficulties could potentially be circumvented by prime editing, a novel CRISPR-based genomic editing approach [20]. Prime editing is mediated by a prime editing guide RNA (pegRNA) and a prime editor, which is a fusion protein consisting of a Cas9 nickase (nCas9) and a reverse transcriptase (RT) [20]. To achieve gene correction via prime editing, the pegRNA first traffics the prime editor near the target site to nick one strand of the genomic DNA, the RT then reverse transcribes an RNA template (with corrected genetic information) embedded in the pegRNA to make a short corrected DNA strand, and this short DNA strand is incorporated into genomic DNA by cellular DNA repair machinery to achieve the correction [20]. Thus, prime editing-mediated gene correction does not require the delivery of a separate DNA repair template. Because of not introducing DSBs, prime editing also minimizes the risk of introducing INDELs. Prime editing has been shown to generate perfect nucleotide replacement up to 80 base pairs [20].

The inefficient delivery of gene correction machinery often represents a major challenge for gene therapy. Due to the large size, it took 2 to 3 lentivirus or adeno-associated viruses (AAVs) to deliver the prime editing machinery into cells [20]. Helper-dependent adenovirus (HDAd) is a promising delivery vehicle in gene therapy with a packaging capacity of up to 37 kb [21], which is sufficient to deliver the entire prime editing machinery. Importantly, HDAd has been demonstrated to efficiently transduce airway cells in cell cultures and animal models [22].

Patient cell-derived induced pluripotent stem cells (iPSCs), which carry all disease-causing mutations and retain the potential to differentiate into every adult cell type, have been considered an alternative human tissue source to model diseases, discover drugs, and develop cell and gene therapies [23]. In this study, we tested whether prime editing can be applied as a potential therapeutic modality to correct nonsense *CFTR* mutations. We generated iPSCs from a CF patient homozygous for the *CFTR* W1282X mutation. We demonstrated that prime editing corrected one mutant allele in iPSCs, which effectively restored CFTR function in iPSC-derived airway epithelial cells and organoids. Furthermore, we demonstrated that iPSC-derived airway epithelial cells may be edited when the prime editing machinery is delivered by HDAd. Together, our study demonstrated that prime editing may be applied to correct *CFTR* mutations such as W1282X.

## Materials and methods

### Ethics statement

Human nasal epithelial (HNE) cells were acquired as described with approval from the IRB of the University of Alabama at Birmingham (Protocol # IRB-151030001) [24]. Written informed

consent, obtained from patients or guardians, was documented, and witnessed by a trained research coordinator.

## Cell culture

HEK 293T cells were obtained from American Type Culture Collection (ATCC). HEK 293T cells were cultured in DMEM (Gibco) with 10% fetal bovine serum (FBS; GeminiBio). iPSCs were maintained and expanded on cell culture plates coated with hESC-qualified Geltrex basement membrane matrix (Thermo Scientific) in mTeSR medium (StemCell Technologies) as described [25].

## iPSC generation and differentiation

iPSCs were generated as described [26]. In brief, 1 x $10^5$ HNE W1282X cells were transduced with the STEMCCA lentivirus [27] and cultured in BEGM supplemented with 10 μM Y-27632 (Cayman Chemicals) for 7 days. The cells were then cultured in the E7 medium for 10 days followed by in the E8 medium for 12 days [28]. Single iPSC colonies were selected and expanded in mTeSR medium (StemCell Technologies) on cell culture plates coated with hESC-qualified Geltrex (Thermo Scientific). iPSCs were differentiated into airway epithelial cells and lung organoids by following the published protocol [29, 30]. In brief, iPSCs were cultured in STEMdiff Definitive Endoderm medium (StemCell Technologies) for 3 days, cSFDM supplemented with 10 μM SB431542 and 2 μM Dorsomorphin (Cayman Chemicals) for 4 days, and cSFDM supplemented with 3 μM CHIR99021 (Cayman Chemicals), 10 ng/mL recombinant human FGF10 (R&D Systems), 10 ng/mL recombinant human KGF (R&D Systems), 10 ng/mL recombinant human BMP4 (PeproTech), 50 nM retinoic acid (Millipore Sigma) for 11 more days. The cells were then mechanically dissociated into cell aggregates, embedded into growth factor-reduced Matrigel drops (50–100 μL in size), and cultured in the cSFDM medium supplemented with 250 ng/mL recombinant human FGF2 (ReproCell), 10 ng/mL FGF10 (R&D Systems), 50 nM dexamethasone (Millipore Sigma), 0.1 mM 8-Bromo-cyclic AMP sodium salt (Cayman Chemicals), 0.1 mM 3-Isobutyl-1-methyl-xanthine (IBMX) (Cayman Chemicals), and 10 μM Y-27632 (Cayman Chemicals) for 7 to 10 days. These cells were used for the differentiation of both airway epithelial cells and lung organoids.

For airway epithelial cell differentiation, single cells digested by Accutase (Thermo Scientific) were plated at 1 x$10^5$ cells/transwell insert of a 24-well plate (Corning), and cultured in PneumaCult-ALI maintenance medium (PAMM, StemCell Technologies) supplemented with 10 μM SB431542, 2 μM Dorsomorphin, and 10 μM Y-27632 for 7 to 12 days until reaching confluency. For air-liquid interface (ALI) culture, the cells were cultured for 2 to 4 weeks with no medium on the apical side. Media in the lower chambers were changed to PAMM medium.

For lung organoid differentiation, cells were dissociated into small aggregates by Accutase, embedded into growth factor-reduced Matrigel drops (50–100 μL in size) (Corning), and cultured in cSFDM medium supplemented with 250 ng/mL FGF2 and 10 ng/mL FGF10 for 7 to 14 days.

## Prime editing vector construction

DNA oligonucleotides for pegRNA and nicking sgRNA (Integrated DNA Technologies) were cloned in pU6-pegRNA-GG-acceptor (Addgene #132777) by Golden Gate assembly (NEB) or the ligation method as described [20]. Sequences of pegRNA and nick RNA are listed in S1 Table in S1 Appendix. PE2 from the pCMV-PE2 plasmid (Addgene #132775) was inserted in pCL195, which drives the PE2-P2A-Puro expression cassette by an EF1α promoter. The pegRNA and nick sgRNA expression cassette was then transferred to pCL195 by Gibson

assembly (NEB) to create the all-in-one pCL195-N-X1282W vector. All vectors were purified using QIAprep Spin Miniprep kits (QIAGEN).

## Prime editing and sib selection

Two million iPSCs were dissociated into single cells by Accutase, washed with DMEM/F12 medium (Gibco), and resuspended in 100 μl Nucleofector solution (Lonza, VPH-5012). 6 μg pCL195-N-X1282W were added into iPSCs containing nucleofector solution, gently mixed, and nucleofected by the A-23 program of a Lonza 2B nucleofector (Lonza). A portion of the cells was directly plated onto a 96-well plate at ~ 50 cells/well for sib selection and the rest cells were plated onto one well of a 6-well plate. All cells were cultured in the mTeSR1 plus medium (StemCell Technologies) supplemented with 10 μM ROCK inhibitor Y-27632 (EMD Milli-pore). 0.375 μg/ml Puromycin was added to the medium 24 hrs later and maintained for 2 days. Genomic DNA samples were prepared by the DNeasy Blood & Tissue Kit (QIAGEN) for ddPCR analyses to estimate editing efficiency. Edited iPSC clones were isolated by sib selection as described [31]. In brief, the cells in the 96-well plate were split into two 96-well plates when the cell reached 80% confluency. One plate was used for genomic DNA extraction and the other plate for continuous cell culture. Genomic DNA was prepared by lysing cells with 25 μl prepGEM Universal (MicroGEM). 1 μl of DNA lysate was used as the template for ddPCR analyses. Wells with the highest gene correction efficiencies were selected for the next round of sib-selection until iPSC clones were isolated.

## Droplet digital PCR (ddPCR) analysis

ddPCR was performed to examine the gene correction efficiency as described [19, 31]. In brief, each reaction contained 11 μl 2x ddPCR Supermix for Probes (Bio-Rad), 2 μl CCR5 primer mix (20 μM), 1 μl CCR5 TaqMan Probe (HEX, 5 μM), 2 μl target primer mix (20 μM), 1 μl tar-get TaqMan Probe (FAM, 5 μM), 100 ng genomic DNA, and water to a final volume of 22 μl. Droplets were prepared by a QX200 Droplet Generator per the manufacturer's instruction (Bio-Rad). PCR reactions were conducted on a Mastercycler Nexus PCR Thermal Cycler (Eppendorf) using a program consisting of 40 cycles of denaturation (95˚C for 30 seconds) and annealing/elongation (57˚C for 1 minute). The droplets were detected and analyzed by a QX200 Droplet Reader (Bio-Rad). Sequences of all primers and probes are listed in S1 Table in S1 Appendix.

## PCR and genotype confirmation by Sanger sequencing

Genomic DNA purified by the DNeasy Blood & Tissue Kit (QIAGEN) was used as the DNA template for PCR reactions to amplify the DNA fragment containing the W1282 codon. The PCR product was purified by DNA Clean & Concentrator Kit (Zymo) before being sequenced using the forward primer. Primer sequences are listed in S1 Table in S1 Appendix.

## Ussing chamber assay

Ussing chamber assay was performed as described [24, 32]. In brief, transwell inserts with iPSC-derived airway epithelial monolayers were mounted in Ussing chambers (Physiologic Instruments, San Diego, CA) and short-circuit currents (Isc) were measured. The cells were initially bathed with Ringers solutions on both sides. After the suppression of the epithelial sodium channel (ENaC) by amiloride (100 μM, apical), the solution on the apical side was replaced by Ringers solution with low chloride to create a chloride gradient as described [32]. Reagents introduced sequentially to the bath solution include 100 μM Amiloride (apical),

10 μM forskolin (apical and basal), 10 μM VX-770 (apical and basal), 10 μM CFTRinh-172 (apical), and 20 μM GlyH101 (apical). Electrophysiological data were analyzed using Acquire and Analyze 2.3 software (Physiologic Instruments, San Diego, CA). All chemicals were purchased from Cayman Chemicals.

## Organoid swelling assay

The organoid swelling assay was performed as described [24]. In brief, lung organoids were seeded onto μ-slide (15-well glass bottom slide, ibidi USA) at a density of ~ 25 organoids/well and cultured in cSFDM medium supplemented with 250 ng/mL FGF2 and 10 ng/mL FGF10 overnight. Right before forskolin stimulation and imaging, the organoids were pre-incubated with NucBlu (Thermo Scientific) for 1 hour. After adding 10 μM forskolin, organoids were imaged every 20 min for 8 hours via the bright field and DAPI fluorescent channels using a Lionheart FX automated imaging system (BioTek). Image processing and the automated quantification of organoid sizes over time were achieved by using the Gen5 ImagePrime software (BioTek).

## Helper-dependent adenoviral (HDAd) vector construction, virus preparation, and HDAd-mediated prime editing

The W1282X correction HDAd vector pCL196 was constructed by transferring the prime editing components from the all-in-one pCL195-N-X1282W vector to pHDAd-26K-GW, an HDAd vector backbone derived from pDelta28ElacZ [21], via I-CeuI/I-SceI. 10 μg of recombinant adenoviral plasmids were digested with PmeI and packaged into HDAd virus by using the helper virus rADNG163 in 116 cells (gifts from Dr. Philip Ng) as described [21, 33]. The titer of the rCL196 HDAd virus was determined by the Endpoint-Dilution Assay [34]. W1282X iPSC-derived airway epithelial cells cultured in ALI culture for 2–3 weeks were used for prime editing. Viral transduction was conducted by adding the rCL196 HDAd virus (an estimated MOI of 100) to the bottom chamber of the transwell. Genomic DNA samples were collected 96 hours after viral transduction and were used for ddPCR analyses to determine editing efficiencies.

## Immunofluorescence and microscopy

Immunofluorescence was conducted as described [35]. In brief, cells were fixed in 4% paraformaldehyde, blocked in Protein Block (Agilent), and incubated with the appropriate primary antibodies overnight at 4°C and secondary antibodies for 1 hr at room temperature. Nuclei were counterstained by 0.5 μg/mL DAPI. F-Actin was stained by phalloidin conjugated with Alexa Fluor 568 (Thermo Scientific). Images were acquired by a Nikon Ti-S or a Nikon A1R-HD25 confocal microscope and processed by Elements AR software (Nikon). Antibodies used were as followings: SOX2 (09–0024, ReproCell), FOXA2 (sc-101060, Santa Cruz Biotech), NKX2.1 (ab76013, Abcam), ZO-1 (MA3-39100-A647, Thermo Scientific), MUC5B (HPA008246, Sigma-Aldrich), α-Tubulin (ab11315, Abcam), CFTR (MAB25031, R&D).

## Results

### *CFTR* W1282X mutation is amenable to prime editing-mediated gene correction

We analyzed the two most common nonsense *CFTR* mutations (i.e., G542X and W1282X) and found the sequence surrounding the W1282X mutation enabled the design of a pegRNA for prime editing (Fig 1A). To experimentally validate that prime editing can correct the

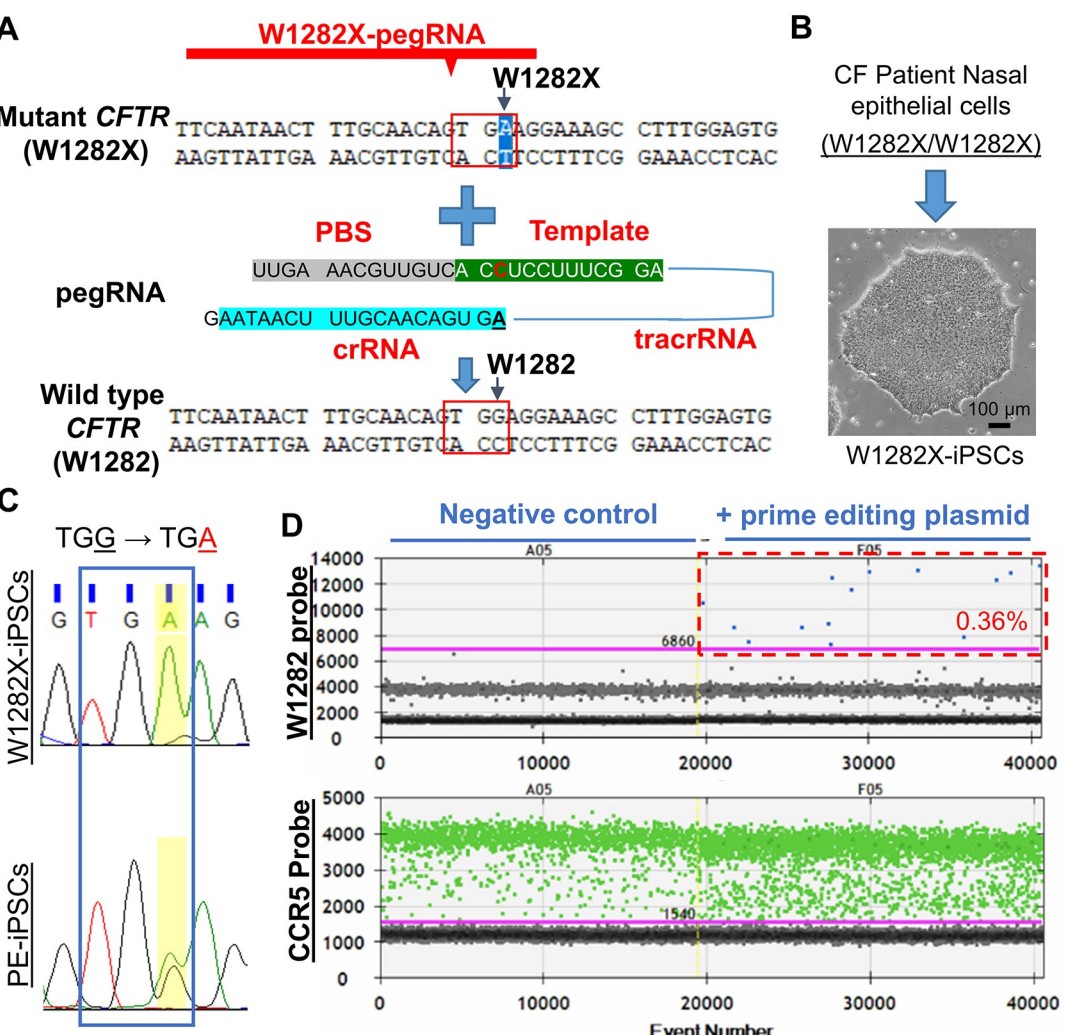

**Fig 1. *CFTR* W1282X mutation is amenable to prime editing-mediated gene correction. (A)** The W1282X-pegRNA and the DNA target. PBS, primer binding site. In addition to the tracrRNA (solid blue line) that binds to the nCas9, each pegRNA contains three key functional components–the crRNA, PBS, and template. crRNA brings the prime editor to the target site. PBS, primer binding site. **(B)** Generation of W1282X-iPSCs from nasal epithelial cells of a CF patient with homozygous *CFTR* W1282X mutations. **(C)** Sequencing analyses of PCR fragments containing the W1282 codon amplified from genomic DNA of W1282X-iPSCs (top) and PE-iPSCs (bottom). Blue box, the W1282X codon. **(D)** Droplet digital PCR (ddPCR) analyses indicated that transfection of prime editing machinery to W1282X-iPSCs corrected 0.36% of mutant alleles. Negative control, W1282X-iPSCs not transfected with prime editing machinery. The W1282 probe only recognizes corrected *CFTR* alleles. The CCR5 probe is used as a copy number control for the input genomic DNA.

W1282X mutation, we first designed a W1282-pegRNA that enables conversion of the wild-type TGG codon encoding the W1282 residue to a TGA stop codon in HEK 293T cells (S1A Fig in S1 Appendix). HEK 293T cells were used to test and optimize prime editing strategies primarily because they are easy to culture and transfect as shown in previous studies [19, 36, 37].

It has been shown that introducing a nick on the non-edited DNA strand by nCas9 may enhance prime editing by facilitating the incorporation of the edited strand during DNA repair [20]. DNA sequence analysis identified two potential nCas9 cutting sites on the non-edited strand (S1B Fig in S1 Appendix). By co-transfecting plasmids expressing the prime editor, W1282-pegRNA, and a nick gRNA, we tested how each of the two nick sites affected the

editing efficiency of the W1282 codon in HEK 293T cells. To determine editing efficiency, we developed droplet digital PCR (ddPCR) assays that can specifically detect the W1282X mutant allele. The ddPCR analysis showed that nick gRNA(-37)-1 led to an approximately 2-fold increase in editing efficiency on the W1282 codon (S1C Fig in S1 Appendix). Based on these data, the nick gRNA(-37)-1 was used for the rest of the study.

The lengths of the primer binding sequence (PBS) and the template of reverse transcription on pegRNAs have also been suggested to impact the efficiency of prime editing in a locus-dependent manner [20]. We then tested editing efficiencies on the W1282 codon by pegRNAs with different PBS and RT template lengths (S1D, S1E Fig in S1 Appendix). We found that pegRNAs with an 11-nucleotide (nt) PBS and a 15-nt RT template were more efficient in editing the W1282 codon (S1F Fig in S1 Appendix). Based on these data, we chose the P11T15 pegRNA, which contains an 11-nt PBS and a 15-nt RT template, to correct the W1282X mutation (Fig 1A).

To test whether prime editing can correct the W1282X mutation and restore CFTR function, we generated iPSCs from a patient homozygous for the *CFTR* W1282X mutation (Fig 1B and 1C). We designed a W1282X-pegRNA targeting the DNA sequence encoding the W1282X mutation (Fig 1A) and transfected the plasmid co-expressing the W1282X-pegRNA, prime editor, and nick gRNA(-37)-1 to W1282X-iPSCs. The corrected *CFTR* alleles can be detected at a frequency of 0.36% by ddPCR analyses, which distinguish the corrected W1282-encoding DNA sequence from the W1282X mutant DNA sequence (Fig 1D). We then performed sib selection to isolate iPSCs carrying prime-editing corrected *CFTR* alleles (PE-iPSCs). We observed a ~10-fold enrichment (0.36% to 3.4%) after one round of selection (S2A Fig in S1 Appendix) and isolated four independent PE-iPSC clones. Sanger sequencing analyses confirmed that all the PE-iPSC clones contained one repaired *CFTR* allele (Fig 1C and S2B Fig in S1 Appendix). Furthermore, INDELs were observed on neither the repaired nor the unrepaired *CFTR* allele in any of the clones (S2B Fig in S1 Appendix).

## Differentiation of iPSCs to airway epithelial cells and lung organoids

Previous studies have demonstrated that iPSC-derived airway epithelial cells and lung organoids may be used to test *CFTR* function in Ussing chamber assays and forskolin-induced swelling assays [29, 38]. To test whether prime editing-mediated gene correction had restored CFTR function, we differentiated iPSCs into airway epithelial cells and lung organoids using the published protocol [29]. iPSC differentiation involves recapitulation of embryonic lung development *in vivo*, which undergoes a series of lineage specification events including definitive endoderm (DE), anterior foregut, lung progenitor cells, and lung epithelium (Fig 2A). We confirmed that the anterior foregut cells expressed the stage-specific transcriptional factors SOX2 and FOXA2 (Fig 2B–2B') and the lung progenitors expressed the stage-specific markers NKX2.1 and FOXA2 (Fig 2C). We also confirmed that the iPSC-derived airway epithelial cells expressed ZO-1, the marker for tight junction, and contained mucin-producing (MUC5B+) and multiciliated (acetylated tubulin+) cells (Fig 2D and 2E). Confocal microscopy further confirmed the apical expression of CFTR in a fraction of airway epithelial cells (Fig 2F).

## Prime editing restored CFTR function in PE-iPSC-derived lung tissues

To test whether prime editing had restored CFTR function, we performed Ussing chamber assays and forskolin-induced swelling (FIS) assays using airway epithelial cells and lung organoids from prime editing-corrected iPSCs (PE-iPSCs) (Fig 3). In the Ussing chamber assays, airway epithelial cells differentiated from PE-iPSCs showed significant activation of short circuit current (Isc) after forskolin (FSK) treatment, which can be further enhanced when

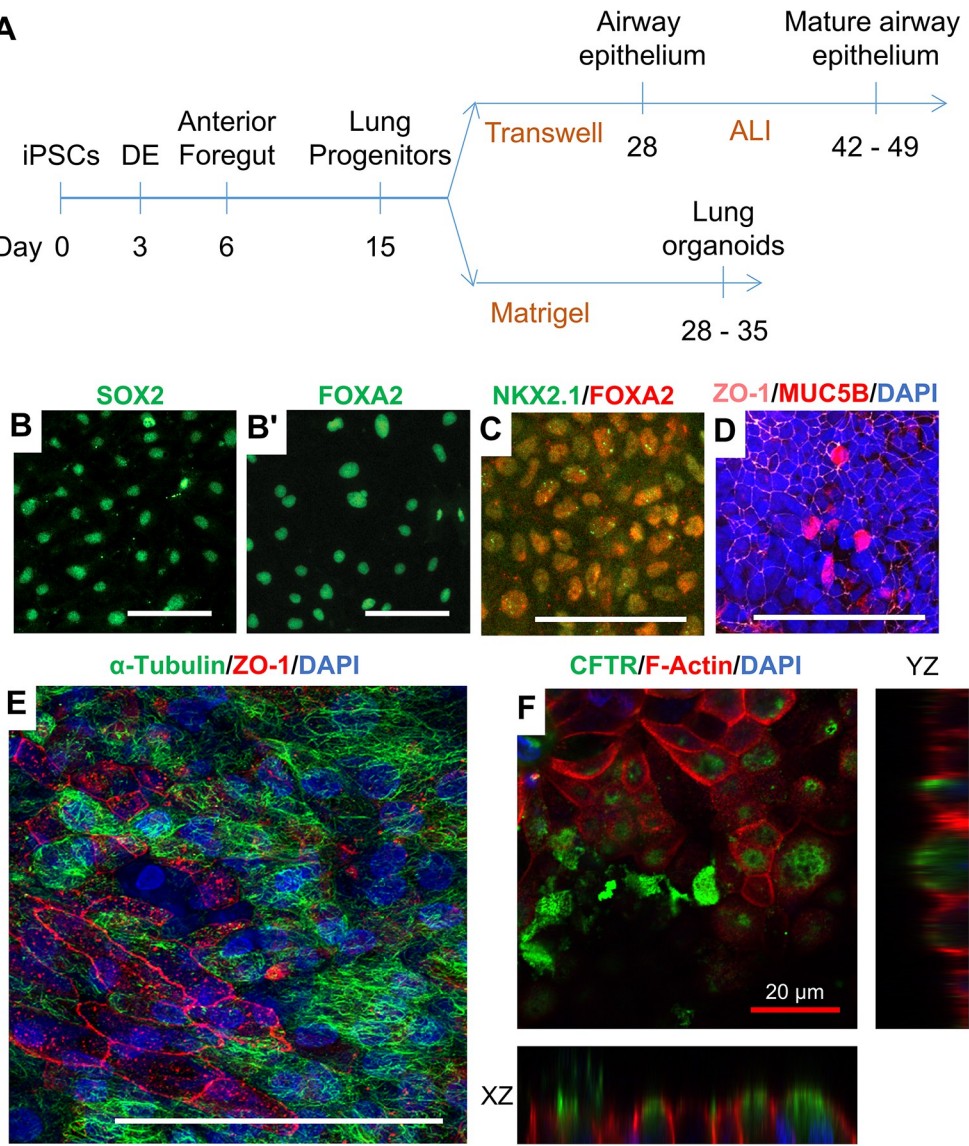

**Fig 2. Differentiation of iPSCs to airway epithelial cells and lung organoids. (A)** Schematic of the iPSC differentiation protocol to airway epithelial cells and lung organoids. DE, definitive endoderm. ALI, air-liquid interface. **(B—F)** Immunofluorescence analyses of stage-specific markers and lung epithelial cell markers. **(B-B')** Anterior foregut cells express stage-specific markers **(B)** SOX2 and **(B')** FOXA2. **(C)** Lung progenitor cells express stage-specific markers NKX2.1 and FOXA2. **(D-F)** iPSC-derived lung epithelium **(D, E)** expresses the tight junction marker ZO-1 and contains cells positive for **(D)** MUB5B, **(E)** acetylated tubulin, and **(F)** CFTR. Note that CFTR is expressed on the apical side (XZ and YZ) of the epithelium. White scale bars, 100 µm. Red scale bar, 20 µm.

treating with VX-770, the potentiator CFTR drug showing pronounced efficacies to mutants with gating defects. The change of Isc depends on the CFTR function because the CFTR inhibitor-172 and GlyH101 can nearly completely suppress the FSK and VX-770-induced current changes (Fig 3A and 3B). In contrast, airway epithelial cells differentiated from W1282X-iPSCs showed responses to neither FSK nor VX-770, supporting that the W1282X mutant is not functional (Fig 3A and 3B). In the FIS assays, lung organoids differentiated from PE-iPSCs swelled upon FSK stimulation over an 8-hour period, while lung organoids differentiated from W1282X-iPSCs exhibited little responses to FSK (Fig 3C and 3D, and supporting movies).

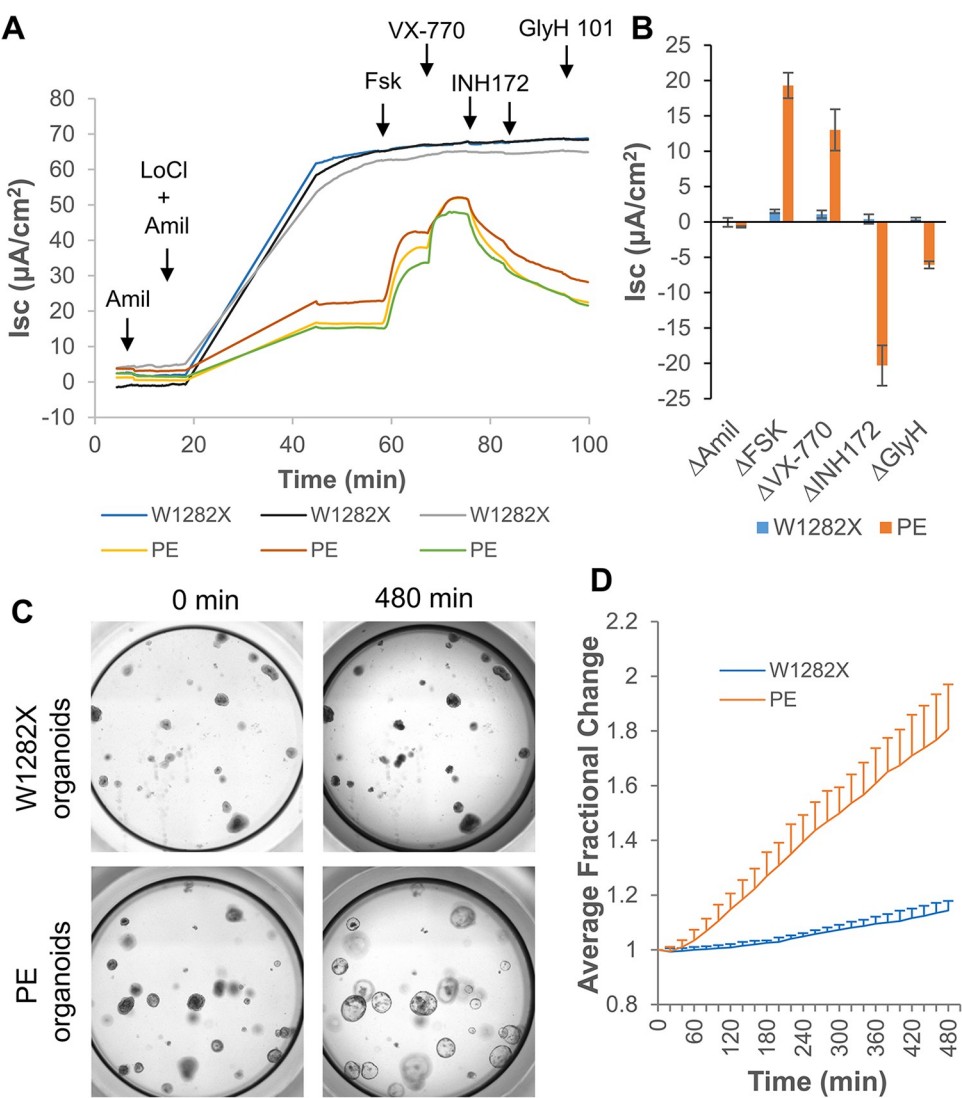

**Fig 3. Prime editing restored CFTR function in PE-iPSC-derived lung tissues. (A)** Ussing chamber assays that measure short circuit current (Isc) traces of airway epithelial differentiated from W1282X-iPSCs and PE-iPSCs. Amil, amiloride; LoCl, low chloride solution; FSK, forskolin; INH 172, CFTR Inhibitor-172; n = 3 biological repeats. **(B)** Quantification of the (A). Error bars, SD. **(C)** Forskolin-induced swelling assays of lung organoids differentiated from W1282X-iPSCs and PE-iPSCs. Shown are images of representative wells before (0 min) and after (480 min) forskolin induction. **(D)** Quantification of (C) was recorded in five independent wells. Error bars, SD.

Together, these data demonstrated that prime editing-mediated gene correction had restored CFTR function.

## Prime editing corrects the W1282X mutation in differentiated airway epithelial cells

To become a potential therapeutic modality, prime editing must correct *CFTR* mutations in differentiated airway epithelial cells but not in undifferentiated iPSCs. Delivery of plasmids that express prime editing machinery to airway epithelial cells by nucleofection and related transfection methods had proven ineffective. To efficiently deliver the prime editing machinery and better mimic a therapeutic setting, we constructed an all-in-one helper-dependent

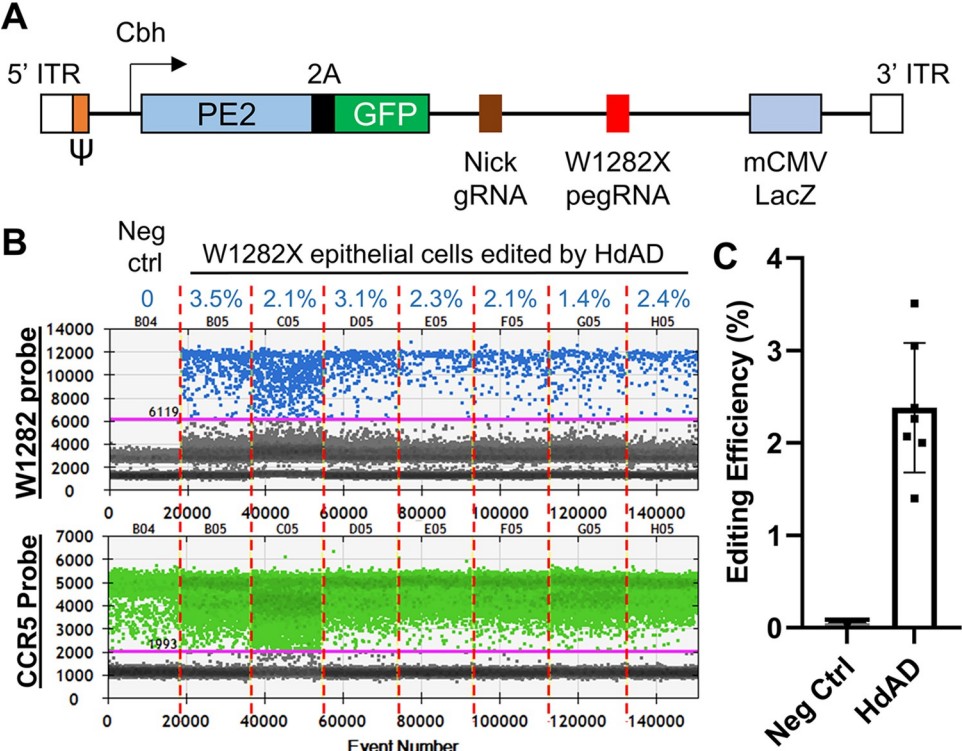

**Fig 4. Prime editing corrects the W1282X mutation in differentiated airway epithelial cells.** (A) Schematic of the all-in-one HDAd vector. (B) ddPCR analyses to measure the correction efficiency in HDAd-transduced airway epithelial cells. Negative control, cells not transduced with HDAd. The W1282 probe only recognizes corrected *CFTR* alleles. The CCR5 probe is used as a copy number control for the input genomic DNA. n = 7 biological repeats. (C) Quantification of (B). Error bars, SD.

adenovirus (HDAd) that expresses all the necessary components of prime editing, which includes the prime editor, the W1282X-pegRNA, the nick gRNA(-37)-1, and a GFP reporter (Fig 4A). Our data showed that HDAd can achieve 71.7 ± 5.4% transduction efficiency in iPSC-derived airway epithelial cells, as monitored by GFP expression (S3 Fig in S1 Appendix). ddPCR analyses demonstrated that 2.4 ± 0.6% of W1282X mutant alleles had been corrected by HDAd-delivered prime editing (Fig 4B and 4C). In the Ussing chamber assays, a minor increase of Isc was observed following FSK stimulation in edited airway epithelia, whereas this response is absent in the non-edited controls (S4A Fig in S1 Appendix). Comparing this to the PE-iPSC-derived airway epithelium (Fig 3B), the amplitude of the FSK-induced Isc is approximately 20-fold lower (S4B Fig in S1 Appendix), which is in line with the expected functional *CFTR* allele frequencies (~2.5% vs. 50%). However, neither the edited nor non-edited airway epithelia exhibited an increase in Isc upon VX-770 stimulation (S4 Fig in S1 Appendix), possibly due to the low Isc signals or other unidentified factors.

## Discussion

Lung diseases are the leading cause of morbidity and mortality in CF patients [1]. Because modeling lung disease requires the use of healthy and diseased human lung epithelial cells, the shortage of human lung tissue hinders CF research and drug discovery. CF patient-derived iPSCs, which carry the disease-causing *CFTR* mutations, have been used to model the disease, discover drug candidates, and test gene therapy strategies. Several laboratories have developed

protocols to differentiate iPSCs into lung tissues [29, 30, 39–46]. The iPSC-derived airway epithelium and lung organoids form tight junctions, express CFTR, and can be used to assay CFTR channel functions by electrophysiological (e.g., whole-cell patch clamp and Ussing chamber assays) and FIS assays [29, 30, 38, 44]. CF patient-derived iPSCs and the derived airway epithelial cells have also been used to test the various genomic editing approaches, including CRISPR/Cas9-mediated HDR [29, 44].

In this study, we demonstrated that prime editing may restore the function of the *CFTR* W1282X mutant. Compared to the conventional CRISPR/Cas9-mediated HDR, prime editing does not generate DSBs, which minimizes the likelihood of introducing INDELs and chromosomal rearrangements. Because prime editing does not rely on homologous recombination, it can edit cells with low proliferative potentials, such as airway epithelial cells (Fig 4). In addition, unlike CRISPR/Cas9-mediated HDR, prime editing does not require a separate repair DNA template. The template module of pegRNA contains the necessary genetic information for gene correction, which is reverse transcribed and incorporated into the genome. Without inducing DSBs and introducing exogenous repair DNA templates, prime editing is likely less toxic to cells.

In this study, we were able to design a pegRNA to correct the *CFTR* W1282X mutation but not the G542X mutation using the currently available prime editor PE2. Because PE2 is derived from the conventional Cas9 protein, which strictly requires the NGG PAM sequence for target recognition [20]. Because Cas9 variants recognizing a broader range of PAM sequences, such as xCas9 or Cas9-NG, had been derived [47, 48], more versatile prime editors that would edit a much larger scope of genomic targets may be developed in future studies to edit genomic targets such as the *CFTR* G542X mutation.

While our data demonstrated prime editing can correct the *CFTR* W1282X mutation in iPSCs and airway epithelial cells, the correction efficiency remains low. The overall correction efficiency is affected by the delivery efficiency of the editing machinery to cells and the editing efficiency of the DNA target. To achieve high delivery efficiency into airway epithelial cells, we used HDAd. Adenovirus is a commonly used delivery vehicle in gene therapy because of its well-characterized biology, broad tropism, and large cloning capacity. Compared to conventional adenovirus, HDAd is less toxic to cells because of the removal of all viral coding sequences [21, 33]. Our data showed that while the great majority of the airway epithelial cells (~ 70%) had received the prime editing machinery (S3 Fig in S1 Appendix), only a fraction (~ 2.5%) of DNA targets were corrected (Fig 4B and 4C). If considering most cells contained a heterozygous repair, the total repaired airway epithelial cells would likely be between 2.5–5%. The editing efficiency of prime editing is locus-dependent. The editing efficiency could be up to 80% at some genetic loci but low at others [49]. Therefore, a better understanding of the variables affecting the editing efficiency, which would lead to further optimization of the prime editing machinery, will be essential to apply prime editing in correcting mutations such as *CFTR* W1282X.

The restoration of 10 to 35% of CFTR activity has been estimated as a necessary threshold to alleviate pulmonary morbidity [50]. Consequently, achieving a correction level spanning from 10 to 35% of mutant *CFTR* alleles is likely imperative for effective CF treatments. However, if the corrected cells primarily comprise basal cells, which can repopulate the airway with functional CFTR-carrying epithelial cells lower editing efficiency may still yield therapeutic benefits. HDAd has proven effective in transducing airway basal cells in animal models [22]. Therefore, gaining a deeper understanding of the specific cell types corrected by prime editing would yield valuable insights into the prospective applications of gene therapy for CF treatment.

Recently, base editing, a novel CRISPR-based genomic editing method that enables an A to G or a C to T transition without introducing DSBs and INDELs [51, 52], had been successfully

applied to repair three *CFTR* mutations including the W1282X mutation [53]. The editing efficiency of the W1282X codon can achieve 69% in cultured cells, which makes base editing among the most promising gene therapy methods. However, base editing generates off-target base substitutions in the DNA genome and particularly on RNA molecules, primarily because base editors can modify any DNA or RNA substrates in proximity [54–58]. The A residue immediately downstream of the *CFTR* W1282X codon had been converted to a G residue in 16% of the genome, which led to a *de novo* arginine to glycine (AGG → GGG) substitution [53]. Compared to base editing, prime editing shows little off-target effects because prime editors only function after sequence complementations between the DNA target and both gRNA and PBS (Fig 1A). However, likely contributed by the additional demand for sequence complementation, prime editing is generally less efficient, and relatively fewer genomic loci are amenable to its modification. Very recently, the *CFTR* F508del mutation had been repaired by prime editing in human intestinal organoids but also at low efficiency [59]. Several recent studies suggested that the efficiency of prime editing may be significantly improved. Nelson et al. reported pegRNAs may be stabilized by adding a structured RNA motif to the 3' ends, which led to a 3 to 4-fold increase in the editing efficiencies of tested loci [60]. Chen et al. reported a 3 to 7-fold increase in prime editing efficiency may be achieved by manipulating the cellular DNA mismatch repair pathway [61]. Anzalone et al. reported a twin prime editing approach, which involves a pair of pegRNAs simultaneously editing two adjacent sites surrounding the targeting locus, could not only enable deletion, replacement, and insertion of larger DNA fragments but also enhance editing efficiency [49]. Together with our results, prime editing may be a strategy to correct *CFTR* mutations, however, further optimization to improve editing efficiency is required.

## Supporting information

**S1 Appendix. It includes four supporting figures, figure legends, and a supporting table.**
(DOCX)

**S2 Appendix. W1282X organoid FIS assay.**
(WMV)

**S3 Appendix. PE organoid FIS assay.**
(WMV)

## Acknowledgments

We thank Dr. Philip Ng for kindly providing adenoviral plasmid, helper virus, and the packing cell line.

## Author Contributions

**Conceptualization:** Chao Li, Zhong Liu, Steven M. Rowe, David M. Bedwell, Jennifer Guimbellot, Rui Zhao.

**Data curation:** Chao Li, Zhong Liu, Justin Anderson, Zhongyu Liu, Liping Tang, Yao Li, Ning Peng, Jianguo Chen, Rui Zhao.

**Formal analysis:** Chao Li, Zhong Liu, Justin Anderson, Zhongyu Liu, Liping Tang, Yao Li, Ning Peng, Jianguo Chen, Xueming Liu, Lianwu Fu, Jennifer Guimbellot, Rui Zhao.

**Funding acquisition:** Rui Zhao.

**Investigation:** Chao Li, Zhong Liu, Justin Anderson, Zhongyu Liu, Rui Zhao.

**Methodology:** Chao Li, Zhong Liu, Justin Anderson, Zhongyu Liu, Liping Tang, Yao Li, Ning Peng, Jianguo Chen, Xueming Liu, Lianwu Fu, Tim M. Townes, Steven M. Rowe, David M. Bedwell, Jennifer Guimbellot, Rui Zhao.

**Project administration:** Rui Zhao.

**Resources:** Rui Zhao.

**Supervision:** Jennifer Guimbellot, Rui Zhao.

**Validation:** Chao Li, Zhong Liu, Justin Anderson, Zhongyu Liu, Rui Zhao.

**Visualization:** David M. Bedwell, Jennifer Guimbellot.

**Writing – original draft:** Chao Li, Zhong Liu, Justin Anderson, Zhongyu Liu, Rui Zhao.

**Writing – review & editing:** Chao Li, Zhong Liu, Rui Zhao.

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
