## [Decision Letter · Decision Letter 0]

4 Sep 2023

PONE-D-23-20582

Prime editing-mediated correction of the CFTR W1282X mutation in iPSCs and derived airway epithelial cells

PLOS ONE

Dear Dr. Zhao,

Thank you for submitting your manuscript to PLOS ONE. After careful consideration, we feel that it has merit but does not fully meet PLOS ONE’s publication criteria as it currently stands. Therefore, we invite you to submit a revised version of the manuscript that addresses the points raised during the review process.

We look forward to receiving your revised manuscript.

Kind regards,

Alessio Branchini

Academic Editor

PLOS ONE

   "R.Z. is supported by Cystic Fibrosis Foundation Research Grant ZHAO19G0, UAB Cystic Fibrosis Research Center Pilot Grant (ROWE15R0), and NIH R01OD026594."

   "R.Z. is supported by Cystic Fibrosis Foundation Research Grant ZHAO19G0, UAB Cystic Fibrosis Research Center Pilot Grant (ROWE15R0), and NIH R01OD026594.

NO - Include this sentence at the end of your statement: The funders had no role in study design, data collection and analysis, decision to publish, or preparation of the manuscript."

   "I have read the journal's policy and the authors of this manuscript have the following competing interests:

S.M.R. provided consulting services and received grants from Novartis, TranslateBio, Galapagos/Abbvie, Synedgen/Synspira, Eloxx, Vertex Pharmaceuticals, Ionis, Astra Zenica, Renovion, Cystetic Medicines, and Arcturus. S.M.R. is the inventor or co-inventor of several patents and held stock or stock options of Synedgen/Synspira and Renovion. "

Reviewers' comments:

Reviewer's Responses to Questions

**Comments to the Author**

1. Is the manuscript technically sound, and do the data support the conclusions?

Reviewer #1: Yes

Reviewer #2: Yes

2. Has the statistical analysis been performed appropriately and rigorously? 

Reviewer #1: No

Reviewer #2: I Don't Know

3. Have the authors made all data underlying the findings in their manuscript fully available?

Reviewer #1: Yes

Reviewer #2: Yes

4. Is the manuscript presented in an intelligible fashion and written in standard English?

Reviewer #1: Yes

Reviewer #2: Yes

5. Review Comments to the Author

Reviewer #1: The study “Prime editing-mediated correction of the CFTR W1282X mutation in iPSCs and derived airway epithelial cells”, by Li and colleagues is of high quality and of interest to a broad audience. The results presented in the main figures of the manuscript provide a concise and impactful study on the use of prime editing to “fix” CFTR mutation p.W1282X. The major advances presented here are the use of an “all-in-one” helper dependent Adenovirus (HDAd) to efficiently deliver all of the prime editing machinery to iPSC-derived airway epithelial cells and restore p.W1282X-CFTR function. Because of the size of the payload, this has previously not been able to be achieved. This reviewer found much of the interesting biology to be in the supplemental figures, where a good amount of optimization was performed. There are only a few comments to address.

Comments:

1) The authors touted the ability of delivering all of the machinery in a helper dependent adenovirus (HDAd) system for efficient editing. However, even with a plasmid system containing all of the PE machinery and high transfection efficiency, the editing efficiency was only 0.36% in iPSCs. (Fig 1D). With the HDAd system providing ~70% transduction efficiency of differentiated airway epithelial cells only corrected 2-3% of the p.W1282X alleles. Indeed, the bottleneck for correction of p.W1282X CF mutations is the overall efficiency of prime editing, not delivery. While there is nice discussion on this issue, the technical limitations of prime editing will take time to resolve. Maybe this bottleneck would not be a significant issue if in that population of transduced cells, basal cells were effectively edited. This would lead to a repopulation of airway epithelial cells with edited alleles and rescued CFTR function, thereby amplifying the therapeutic impact of low-efficiency prime editing. Important for this study, what cell types in the differentiated airway epithelia were transduced by HDAd? Are basal cells included in this population?

2) Why was functional rescue of p.W1282X-CFTR prime edited differentiated epithelial cells not reported? It would seem that even with as low as 1.5% editing there would be measurable rescue of CFTR function by Ussing chamber assays for measurement of short circuit current (Isc). Indeed, the CFTR functional rescue is often significantly higher than the genome editing efficiency.

Reviewer #2: I have reviewed the manuscript numbered PONE-D-23-20582, titled “Prime editing-mediated correction of the CFTR W1282X mutation in iPSCs and derived airway epithelial cells”, submitted by Chao L., Zhong L., et al.

This article, presenting an original research, is very interesting and very clearly described.

In the manuscript, the authors have developed a novel PRIME editing system for the correction of the W1282X CFTR mutation.

The authors did a very good initial screening job for the components of the PRIME editing System (as reported also in supplementary information) and then, validated the best one in iPSCs generated from nasal epithelial cells of a homozygous W1282X cystic fibrosis patient.

PE-iPSCs were also differentiated into airway epithelial cells and lung organoids, for testing the restored CFTR function. As a very important point to become a therapeutic modality, the Authors tested the PRIME editing system directly in differentiated airway epithelial cells, using an HDAd vector containing all the sequences needed for the prime editing system (all-in –one vector).

In conclusion, the Authors demonstrated the correction of the point mutation albeit in a limited number (2.5-5%) of iPS-derived epithelial cells and, very importantly, getting no indels effect though the DNA.

These results could be very useful for the development of an in-vivo PRIME editing system for stop mutation in CF.

The only suggestion to the Authors could be to add a sentence in the discussion, in particular in the paragraph stating the editing efficiency, indicating (if it is known) what could hypothetically be the correction rate to be reached by PRIME-CRISPR/Cas9-based systems, in order to get a therapeutic level of CFTR for CF patients.

Other specific comments:

-Are Results and Conclusions presented in an appropriate fashion and are supported by the data?

YES. Results are very well written, linear, and complete. The same is for the Conclusions (see my only suggestion above).

-Is the article presented in an intelligible fashion and written in standard English?

YES

- Does the research meet all applicable standards for the ethics of experimentation and research integrity?

YES

6. PLOS authors have the option to publish the peer review history of their article (what does this mean?). If published, this will include your full peer review and any attached files.

Reviewer #1: No

Reviewer #2: No

---

## [Author Response · Author response to Decision Letter 0]

18 Oct 2023

We sincerely thank the editor’s comments and feedback. We have updated and formatted the revised manuscript accordingly. Below are our point-to-point responses. 

We formatted the revised manuscript according to the style requirements. This includes changing level 1 and 2 headings to appropriate fonts, abiding by the requirements for figure citations/captions, and updating the style of supporting information citations and acknowledgments. We also updated the title, author, and affiliation style to abide by the guidelines. 

 "R.Z. is supported by Cystic Fibrosis Foundation Research Grant ZHAO19G0, UAB Cystic Fibrosis Research Center Pilot Grant (ROWE15R0), and NIH R01OD026594."

We note that you have provided funding information that is not currently declared in your Funding Statement. 

We have updated the funding information in the revised submission.

However, funding information should not appear in the Acknowledgments section or other areas of your manuscript. We will only publish funding information present in the Funding Statement section of the online submission form. 

 "R.Z. is supported by Cystic Fibrosis Foundation Research Grant ZHAO19G0, UAB Cystic Fibrosis Research Center Pilot Grant (ROWE15R0), and NIH R01OD026594.

Funding information has been deleted from the Acknowledgements section. 

NO - Include this sentence at the end of your statement: The funders had no role in study design, data collection and analysis, decision to publish, or preparation of the manuscript."

The below statement has been included in the cover letter.

“The funders had no role in study design, data collection and analysis, decision to publish, or preparation of the manuscript."

 "I have read the journal's policy and the authors of this manuscript have the following competing interests:

S.M.R. provided consulting services and received grants from Novartis, TranslateBio, Galapagos/Abbvie, Synedgen/Synspira, Eloxx, Vertex Pharmaceuticals, Ionis, Astra Zenica, Renovion, Cystetic Medicines, and Arcturus. S.M.R. is the inventor or co-inventor of several patents and held stock or stock options of Synedgen/Synspira and Renovion. "

The below statement has been included in the cover letter and revised manuscript.

“This does not alter our adherence to PLOS ONE policies on sharing data and materials. "

“data not shown” has been removed from the revised manuscript because it is not a core part of the reported study. 

Captions for Supporting Information have been added to the end of the revised manuscript to meet the guidelines.

 

We sincerely thank the reviewers for their positive comments and critical feedback, which we have used to revise and strengthen our manuscript. Please find the below point-to-point responses. We anticipate that our study on the correction of CFTR W1282X mutation by prime editing will be of great interest to readers of PLOS ONE.

Reviewer #1: The study “Prime editing-mediated correction of the CFTR W1282X mutation in iPSCs and derived airway epithelial cells”, by Li and colleagues is of high quality and of interest to a broad audience. The results presented in the main figures of the manuscript provide a concise and impactful study on the use of prime editing to “fix” CFTR mutation p.W1282X. The major advances presented here are the use of an “all-in-one” helper dependent Adenovirus (HDAd) to efficiently deliver all of the prime editing machinery to iPSC-derived airway epithelial cells and restore p.W1282X-CFTR function. Because of the size of the payload, this has previously not been able to be achieved. This reviewer found much of the interesting biology to be in the supplemental figures, where a good amount of optimization was performed. There are only a few comments to address.

Comments:

1) The authors touted the ability of delivering all of the machinery in a helper dependent adenovirus (HDAd) system for efficient editing. However, even with a plasmid system containing all of the PE machinery and high transfection efficiency, the editing efficiency was only 0.36% in iPSCs. (Fig 1D). With the HDAd system providing ~70% transduction efficiency of differentiated airway epithelial cells only corrected 2-3% of the p.W1282X alleles. Indeed, the bottleneck for correction of p.W1282X CF mutations is the overall efficiency of prime editing, not delivery. While there is nice discussion on this issue, the technical limitations of prime editing will take time to resolve. Maybe this bottleneck would not be a significant issue if in that population of transduced cells, basal cells were effectively edited. This would lead to a repopulation of airway epithelial cells with edited alleles and rescued CFTR function, thereby amplifying the therapeutic impact of low-efficiency prime editing. Important for this study, what cell types in the differentiated airway epithelia were transduced by HDAd? Are basal cells included in this population?

We agree that the observed low correction efficiency could hold therapeutic significance if the corrected cells are predominantly basal cells. For iPSC differentiation into airway epithelium, we primarily followed the published protocol (ref. 29 and 30), which demonstrated the presence of basal cells in iPSC-derived epithelium. HDAd can effectively transduce airway basal cells in animal models (ref. 22). However, determining prime editing-corrected cell types is technically challenging. Our current experimental system detects the prime editing-mediated single nucleotide substitution solely by ddPCR. The ddPCR procedure prevents simultaneous cell type identification. 

A potential solution to this important question is to construct a W1282X reporter iPSC line, in which a GFP reporter will be fused in-frame to the 3’ end of the mutant CFTR W1282X open reading frame. Because of the W1282X mutation, uncorrected airway cells would remain GFP negative but airway cells with correction would become GFP positive. The GFP-positive cells can then be FACS-sorted for marker staining and single-cell sequencing analysis. However, this approach involves a significant amount of efforts and we believe it should become a follow-up study of the current report. We agree this is an important point that should be further explored. We have included a discussion on the potential implications of correcting basal cells in the revised manuscript (Page 15).

2) Why was functional rescue of p.W1282X-CFTR prime edited differentiated epithelial cells not reported? It would seem that even with as low as 1.5% editing there would be measurable rescue of CFTR function by Ussing chamber assays for measurement of short circuit current (Isc). Indeed, the CFTR functional rescue is often significantly higher than the genome editing efficiency.

We agree with the reviewer and have conducted the recommended experiment (supplemental Figure S4). In the prime-edited airway epithelium, we detected a minor Isc spike following FSK stimulation, a response absent in the unedited controls. When compared to the PE-iPSC-derived airway epithelium (Fig. 3B), the amplitude of the FSK-induced Isc is approximately 20-fold lower, which is in line with the expected functional CFTR allele frequencies (~2.5% vs. 50%). However, we did not observe a VX-770-induced Isc change, which could be attributed to the generally low Isc signals of the HDAd-corrected epithelium or other unidentified factors. We have included these new data in the revised manuscript (Fig. S4 and page 13). 

Reviewer #2: I have reviewed the manuscript numbered PONE-D-23-20582, titled “Prime editing-mediated correction of the CFTR W1282X mutation in iPSCs and derived airway epithelial cells”, submitted by Chao L., Zhong L., et al.

This article, presenting an original research, is very interesting and very clearly described.

In the manuscript, the authors have developed a novel PRIME editing system for the correction of the W1282X CFTR mutation. The authors did a very good initial screening job for the components of the PRIME editing System (as reported also in supplementary information) and then, validated the best one in iPSCs generated from nasal epithelial cells of a homozygous W1282X cystic fibrosis patient. PE-iPSCs were also differentiated into airway epithelial cells and lung organoids, for testing the restored CFTR function. As a very important point to become a therapeutic modality, the Authors tested the PRIME editing system directly in differentiated airway epithelial cells, using an HDAd vector containing all the sequences needed for the prime editing system (all-in –one vector).

In conclusion, the Authors demonstrated the correction of the point mutation albeit in a limited number (2.5-5%) of iPS-derived epithelial cells and, very importantly, getting no indels effect though the DNA.

These results could be very useful for the development of an in-vivo PRIME editing system for stop mutation in CF.

The only suggestion to the Authors could be to add a sentence in the discussion, in particular in the paragraph stating the editing efficiency, indicating (if it is known) what could hypothetically be the correction rate to be reached by PRIME-CRISPR/Cas9-based systems, in order to get a therapeutic level of CFTR for CF patients.

We agree that this is an important point. It is believed that restoration of 10 to 35% of CFTR activities would be required to mitigate pulmonary morbidity (PMID: 15510065). Based on this estimation, a correction level ranging from 10 to 35% mutant alleles would be necessary. However, if the corrected cells primarily comprise basal cells, therapeutic advantages could potentially be attained with lower editing efficiency. We have included this discussion in the revised manuscript (page 15).

---

## [Decision Letter · Decision Letter 1]

14 Nov 2023

Prime editing-mediated correction of the CFTR W1282X mutation in iPSCs and derived airway epithelial cells

PONE-D-23-20582R1

Dear Dr. Zhao,

We’re pleased to inform you that your manuscript has been judged scientifically suitable for publication and will be formally accepted for publication once it meets all outstanding technical requirements.

Kind regards,

Alessio Branchini

Academic Editor

PLOS ONE

Reviewers' comments:

Reviewer's Responses to Questions

**Comments to the Author**

1. If the authors have adequately addressed your comments raised in a previous round of review and you feel that this manuscript is now acceptable for publication, you may indicate that here to bypass the “Comments to the Author” section, enter your conflict of interest statement in the “Confidential to Editor” section, and submit your "Accept" recommendation.

Reviewer #1: All comments have been addressed

2. Is the manuscript technically sound, and do the data support the conclusions?

Reviewer #1: Yes

3. Has the statistical analysis been performed appropriately and rigorously? 

Reviewer #1: Yes

4. Have the authors made all data underlying the findings in their manuscript fully available?

Reviewer #1: Yes

5. Is the manuscript presented in an intelligible fashion and written in standard English?

Reviewer #1: Yes

6. Review Comments to the Author

Reviewer #1: (No Response)

7. PLOS authors have the option to publish the peer review history of their article (what does this mean?). If published, this will include your full peer review and any attached files.

Reviewer #1: **Yes: **John D. Lueck

---

## [Editor Report · Acceptance letter]

17 Nov 2023

PONE-D-23-20582R1 

Prime editing-mediated correction of the *CFTR* W1282X mutation in iPSCs and derived airway epithelial cells 

Dear Dr. Zhao:

I'm pleased to inform you that your manuscript has been deemed suitable for publication in PLOS ONE. Congratulations! Your manuscript is now with our production department. 

Kind regards, 

on behalf of

Dr. Alessio Branchini 

Academic Editor

PLOS ONE